# The Indo-Pacific Pollen Database - a Neotoma constituent database

Annika V. Herbert[1,2,3], Simon G. Haberle[1,2,3], Suzette G.A. Flantua[4], Ondrej Mottl[4,5,6], Jessica L. Blois[7], John W. Williams[8], Adrian George[8], Geoff Hope[1†]

[1]Department of Archaeology and Natural History, Australian National University, Canberra, ACT 2601, Australia.

[2]Australian Research Council Centre of Excellence in Australian Biodiversity and Heritage, Australian National University, Canberra, ACT 2601, Australia.

[3]Australian Research Council Centre of Excellence for Indigenous and Environmental Histories and Futures, Australian National University, Canberra, ACT 2601, Australia.

[4]Department of Biological Sciences, University of Bergen, Bjerknes Centre for Climate Research, N-5020 Bergen, Norway.

[5]Center for Theoretical Study, Charles University, Jilská 1, 11000-CZ Praha 1, Czech Republic

[6]Department of Botany, Faculty of Science, Charles University, Benátská 2, CZ-12801 Prague, Czech Republic

[7]Department of Life and Environmental Sciences, University of California-Merced, Merced, CA 95343, USA.

[8]Department of Geography and Center for Climatic Research, University of Wisconsin, Madison, WI 53706, USA.

†Deceased

*Correspondence to:* Annika V. Herbert (annika.herbert@anu.edu.au)

**Abstract.** The Indo-Pacific Pollen Database (IPPD) is the brainchild of the late Professor Geoffrey Hope, who gathered pollen records from across the region to ensure their preservation for future generations of palaeoecologists. This noble aim is now being fulfilled by integrating the IPPD into the online Neotoma Palaeoecology Database, making this compilation available for public use. Here we explore the database in depth and suggest directions for future research. The IPPD comprises 226 fossil pollen records, most postdating 20 ka, but some extending as far back as 50 ka or further. Over 80% of the records are Australian, with a fairly even distribution between the different Australian geographical regions, the notable exception being Western Australia, which is only represented by 3 records. The records are also well distributed in modern climate space, the largest gap being in drier regions due to preservation issues. However, many of the records contain few samples or have fewer than 5 chronology control points, such as radiocarbon, luminescence or Pb-210 for the younger sequences. Average deposition time for the whole database,

counted as years per cm, is 64.8 yr/cm, with 61% of the records having a deposition time of less than 50 yr/cm. The slowest deposition time by geographical region occurs on Australia's east coast, while the fastest times are from the Western Pacific. Overall, Australia has a slower deposition time than the rest of the Indo-Pacific region. The IPPD offers many exciting research opportunities to investigate past regional vegetation changes and associated drivers, including contrasting the impact of first human arrival and European colonisation on vegetation. Examining spatio-temporal patterns of diversity and compositional turnover/rate of change, land cover reconstructions, plant functional or trait diversity are other avenues of potential research, amongst many others. Merging the IPPD into Neotoma also facilitates inclusion of data from the Indo-Pacific region into global syntheses.

## 1. Introduction

The digital revolution and the advent of the internet transformed the landscape of fossil sample databases, enabling them to be shared with a global audience and heralding an era of scientific transparency and cooperation. The Neotoma Palaeoecology Database represents a significant stride in this direction, offering a diverse range of records including fossil pollen, charcoal, vertebrates, diatoms, ostracods, insects and geochronological data (Williams *et al.,* 2018). This great resource has been used in hundreds of scientific publications to date, thus enabling the science of palaeoecology to take significant steps forward. In the era of open science, the importance of making scientific data publicly available cannot be overstated. Open access to data not only fosters transparency in research but also enables collaboration and innovation across disciplines and geographical boundaries (Wolkovich et al., 2012; Record et al., 2022). This approach is crucial in the field of palaeoecology, where comprehensive and accessible databases are key to understanding ecological histories and predicting future environmental changes (e.g. Lyver *et al.,* 2015).

The journey of large databases in capturing fossil data spans several decades, beginning with pioneering efforts like Newell's palaeontological database in 1952. The advent of spatial analyses of fossil data gave rise to significant developments, such as the European, North American and Latin American Pollen Databases, which were integral to the concept of a Global Pollen Database, all of which are now constituent databases in Neotoma (Gajewski, 2008; Whitmore et al., 2005; Fyfe et al., 2009; Flantua et al., 2015). A notable extension of this endeavour was the Indo-Pacific Pollen Database (IPPD), originally compiled as part of

the BIOME 6000 project. The aim of this project was to create biome maps for various periods (Prentice and Webb, 1998; Pickett et al., 2004), namely the present day, the mid-Holocene, defined as 6,000 ± 500 cal yr BP and the Last Glacial Maximum (LGM), defined as 21,000 ± 1,000 cal yr BP (Pickett *et al.,* 2004). Subsequent enhancements involved the inclusion of full records, as opposed to time slices, with a focus on Australian records (Herbert and Harrison, 2016), the full incorporation of an Indo-Pacific compilation done by the late Professor Geoff Hope (Hope et al., 1999), and modern samples (<100 yr BP). The latter served as a training data set for palaeoclimate reconstructions for comparison with palaeoclimate models (Herbert and Harrison, 2016). This version of the database has been used in multiple wide-ranging studies to date, including global overviews of rates of change processes (Mottl et al., 2021) and pollen taxonomic harmonisation processes (Birks *et al.,* 2023). In addition, it has been used in regional studies, examining the climate dynamics of the last glacial period (Cadd *et al.,* 2021; Herbert and Fitchett, 2021), the importance of Indigenous landscape and fire management (Mariani *et al.,* 2022), biodiversity dynamics (Adeleye *et al.,* 2021) and plant functional dynamics (Adeleye *et al.,* 2023). This has added to the wealth of palaeoecological studies from this diverse region, such as Peter Kershaw's work on a southeast Australian pollen database (Kershaw *et al.,* 1994; D'Costa and Kershaw, 1997). This is separate from the IPPD, but related, and contains many of the same sites, though with a focus on pre-European samples. Examples of other important work in this region that used their own pollen sample compilations is a floristic diversity study of South Pacific Islands (Strandberg *et al.,* 2024) and a study of human impact on the biodiversity of islands (Nogué *et al.,* 2021).

Combined with the recent addition to Neotoma of other sites in the Southern Hemisphere through both the Latin American Pollen database (including South American sites, Flantua *et al.,* 2015) and the African Pollen Database (Ivory *et al.,* 2020; Lézine *et al.,* 2021) (both constituent databases in Neotoma), the addition of the IPPD to Neotoma suggests the enticing prospect of truly global palaeoecological studies. Here we explore this new constituent database in depth and suggest directions for future research.

## 2. Methods

Due to the recognised importance of openly accessible palaeoecological databases, fully integrating the IPPD into Neotoma (https://www.neotomadb.org/) has been a long-term goal.

Since 2021 a concerted effort has been made to fulfil this goal, which has involved getting hundreds of records in the right format and double-checking all the data and metadata against available publications, as well as removing duplicated entries and other data entry errors. Due to the original data no longer being available, some records had to be digitised from publications or theses. When this was done, the digitisation process and quality of results were carefully checked. Upon completing the digitisation, the stated pollen sum had to be 100 ± 10% for the record to be accepted. Being such an underrepresented region, the integration also involved adding over 400 pollen taxa to Neotoma's taxonomic structure. This is a complicated procedure, as every single taxon needs to be validated against up-to-date and trusted sources. For most of these taxa, we used The Australian Plant Name Index (https://biodiversity.org.au/nsl/services/search/names), and references therein.

Age models for all records have been reconstructed using Bacon (Blaauw and Christen, 2011) in R (R Core Team, 2022), versions depending on when the age model was constructed, as the database has been updated in stages. Radiocarbon dates were calibrated using either SHCal20 (Hogg et al., 2020) or SHCal13 (Hogg et al., 2013), again depending on when the age model was constructed.

For ease of presentation, we will here only present fossil records with more than 2 chronological control points and at least 3 stratigraphic levels, thus excluding many surface sample collections. The minimum number of chronological control points was chosen to be able to construct robust age depth models. Chronological control points include dates obtained through radiocarbon dating, luminescence dating (either thermally or optically stimulated), U/Th, or Pb-210 for younger sequences. Plots were produced in R (R Core Team, 2022) to show different aspects of the IPPD (see supplementary information).

Modern observational climate data was taken from the CRU TS v4.07 gridded dataset at 0.5° grid cell resolution (Harris et al., 2020). The location of each record in the IPPD was then plotted against the full modern dataset, with the chosen climate variables being: Mean Annual Precipitation (MAP), Mean Annual Temperature (MAT), Mean Temperature of the Coldest month (MTCO) and Mean Temperature of the Warmest Month (MTWA).

# 3. Results and discussion

The IPPD holds records from a total of 530 sites, many of which contain only surface samples. A total of 226 stratigraphic records covering the Indo-Pacific region are presented here, which excludes surface samples and poorly dated records. These 226 records contain 9765 samples, with an average of 43 samples per record (SD: 51.3). There are 2461 unique taxa in the database, with an average of 56 taxa per record (SD: 31.9).

## 3.1 Spatial distribution

Over 83% of the records are from Australia, with high representation from each Australian region, apart from Western Australia (Fig. 1a). Most pollen samples available through the IPPD consist of raw counts (58.4%, Fig. 1b), the rest are percentages, as well as a very small number of concentration counts. Just over a quarter of the records (27%) have had to be digitised from publications or theses, due to the raw data no longer being available. Every region in the IPPD is represented mostly by sequences containing raw counts, which can therefore be used to verify the quality of the sequences from digitised sources from the same region (Figs. 1b, A2). There is also a good representation in modern climate space, with the IPPD sites covering most of the available observed climate space, the largest gap being in the driest regions due to preservation issues (Fig. 2). While there is a high number of sites from the southeastern coast of Australia (Appendix A, Fig. A1), the sites in the IPPD cover much of the available modern climate space despite this spatial bias (Fig. 2). This is invaluable when performing palaeoclimate reconstructions, as most methods use some form of space-for-time calibration that relies on networks of surface samples crossreferenced with current climate conditions (Herbert and Harrison, 2016; Chevalier *et al.,* 2020).

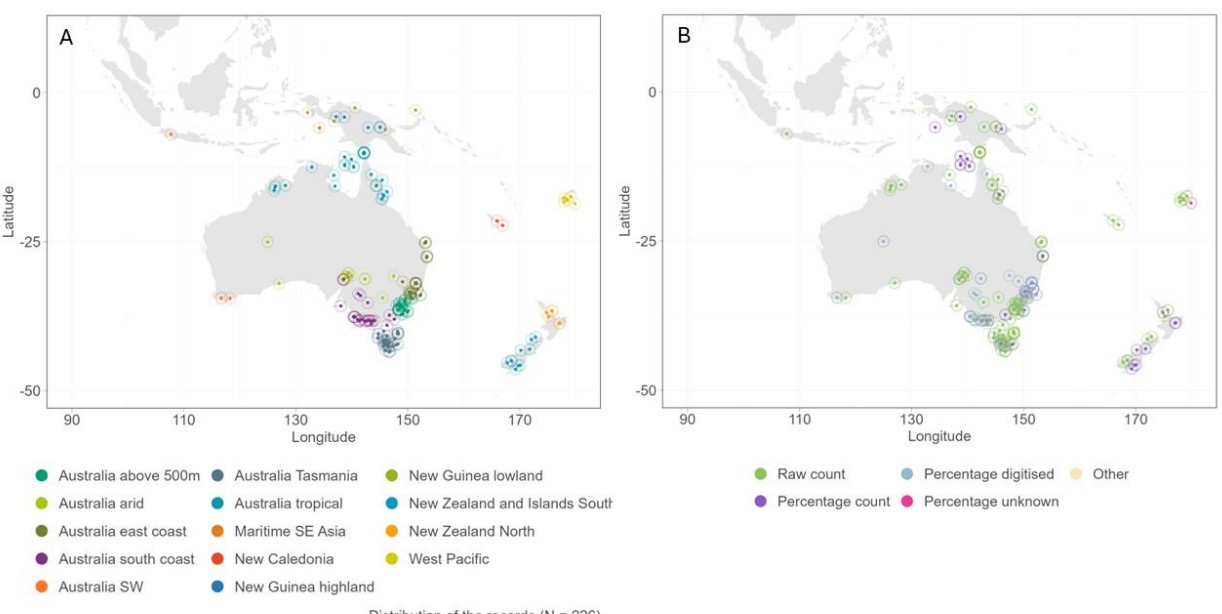

Distribution of the records (N = 226)

**Figure 1,** a) Site locations by geographical region, b) geographical distribution of records by count type, where "Other" refers to concentration data, and "Percentage unknown" may or may not have been digitised. For detailed histograms, see Figs. A1 and A2 in the Appendix.

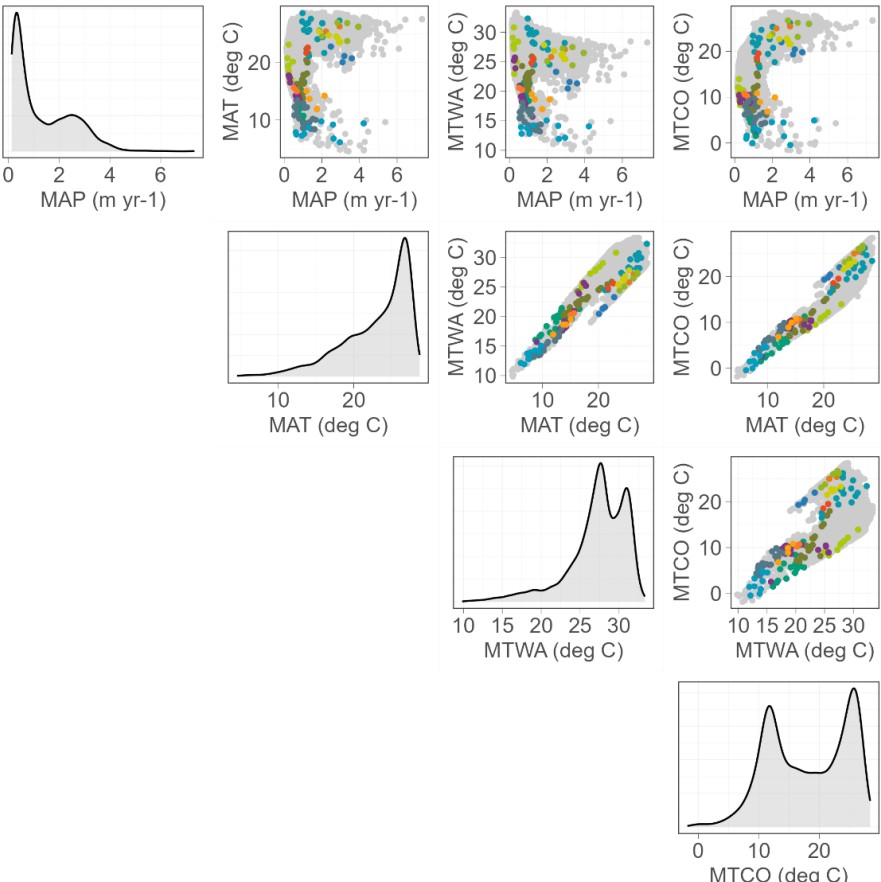

**Figure 2,** Modern climate space covered by sites in the IPPD. Available observed climate space for the Indo-Pacific region in grey (from CRU TS v4.07, Harris *et al.,* 2020), IPPD sites coloured according to geographical region, as presented in Fig. 1a. The line diagrams show the distribution of each climate variable, where MAP = Mean Annual Precipitation, MAT = Mean Annual Temperature, MTCO = Mean Temperature of the Coldest Month, MTWA = Mean Temperature of the Warmest Month.

*3.2 Depositional environment*

There are a total of 33 different depositional environments represented in the database, with lakes and wetlands being the main source of records (73% of sites, Fig. 3). 31% of the records come from lakes, natural or man-made, with a further 42% coming from wetlands, such as swamps or fens.

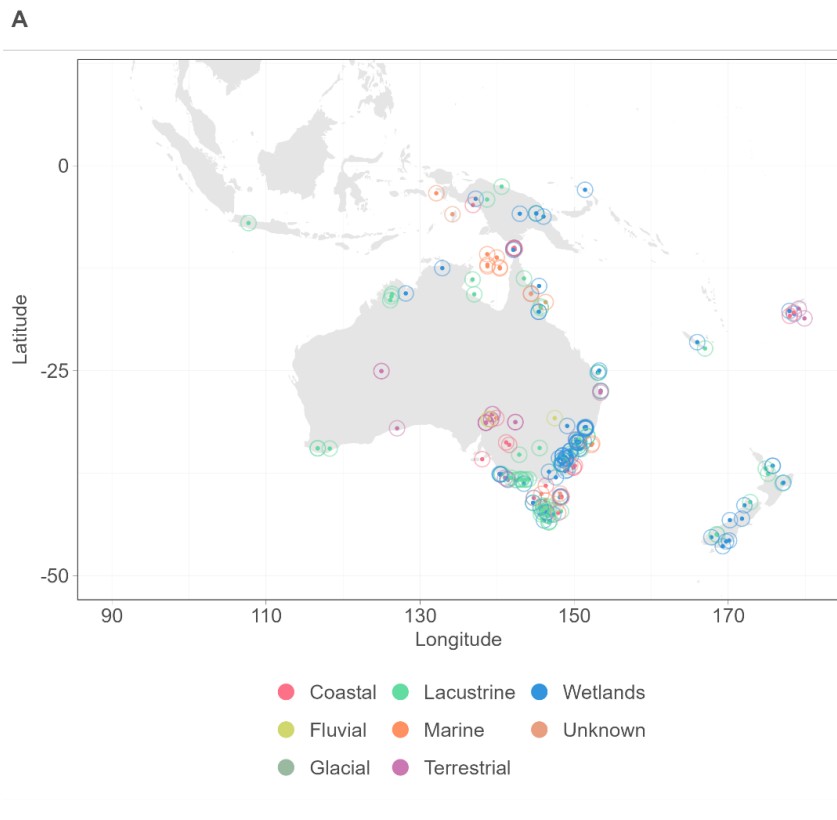

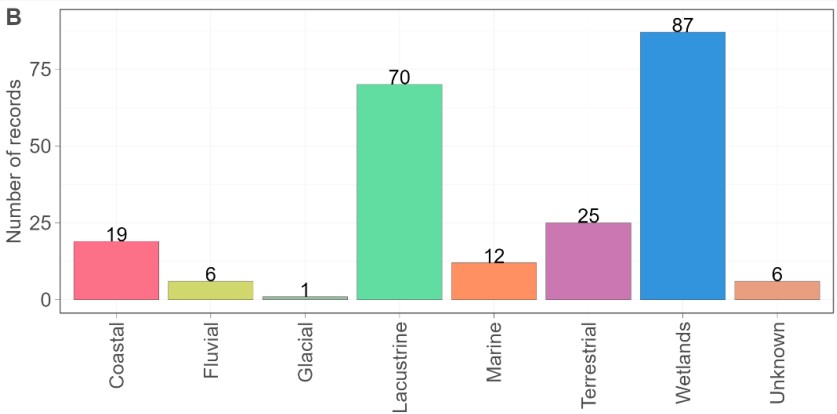

**Figure 3,** a) geographical distribution of records by depositional environment; b) number of records by depositional environment, binned into broader groupings. For detailed histograms, see Fig. A3 in the Appendix. For a full list of depositional environments, see Supplementary Table S1.

## *3.3 Temporal distribution and chronological control*

There are a total of 1637 chronological control points in the database, with an average of 7 points per record (SD: 7.8). 52.6% of the records contain fewer than 5 chronological points each, meaning they may be poorly dated (Fig. 4a), depending on the number of levels associated with the site (Fig. 4b).

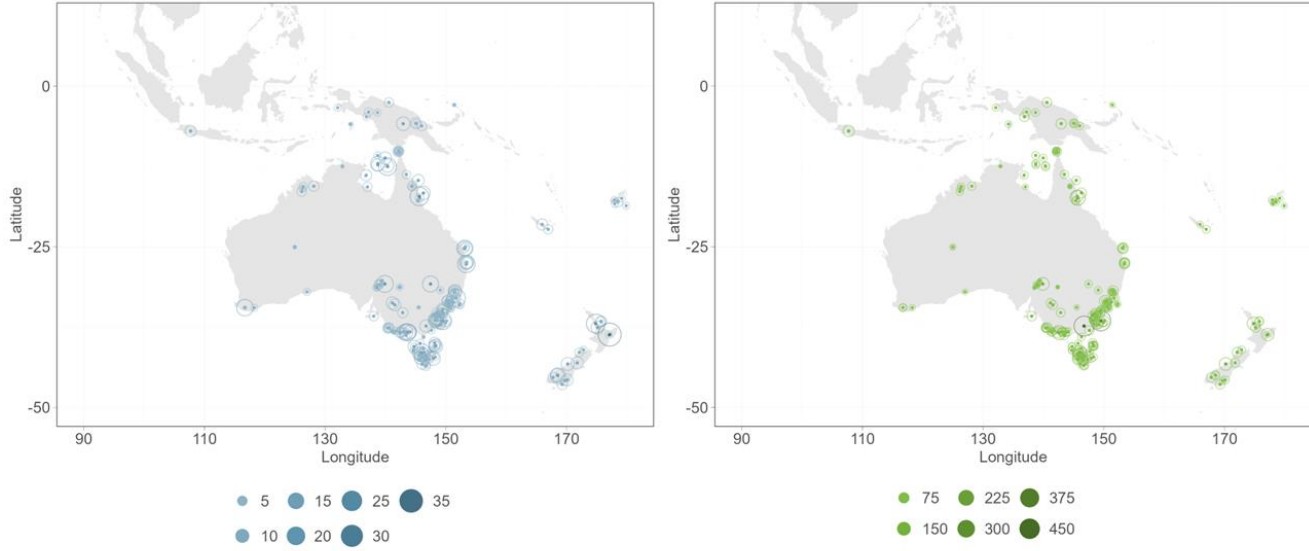

**Figure 4,** a) Geographical distribution of sites by number of chronological control points, b) geographical distribution of sites by number of levels. For a detailed histogram, see Fig. A4 and A5 in the Appendix.

Some records (17 out of 226, e.g. 7.5%) in the IPPD date back more than 50 ka BP, but most are younger than 20 ka (180 out of 226, e.g. 79.6%). In addition, many sequences are not continuous and are composed of a few levels at the older end (Fig. 5). No clear geographical pattern in the distribution of short sequences or sequences with fewer than 5 chronological control points can be discerned from Fig. 4, meaning the length of the sequences is not necessarily climate dependent. However, most of the older sequences are located close to the coast where they are more likely to receive high rainfall, keeping the sediments wet and preserving the pollen grains (Fig. 5). Keeping the sediments waterlogged also decreases the risk of erosion, thereby benefiting the development of long sequences of well-preserved pollen (Delcourt and Delcourt, 1980; Lowe, 1982).

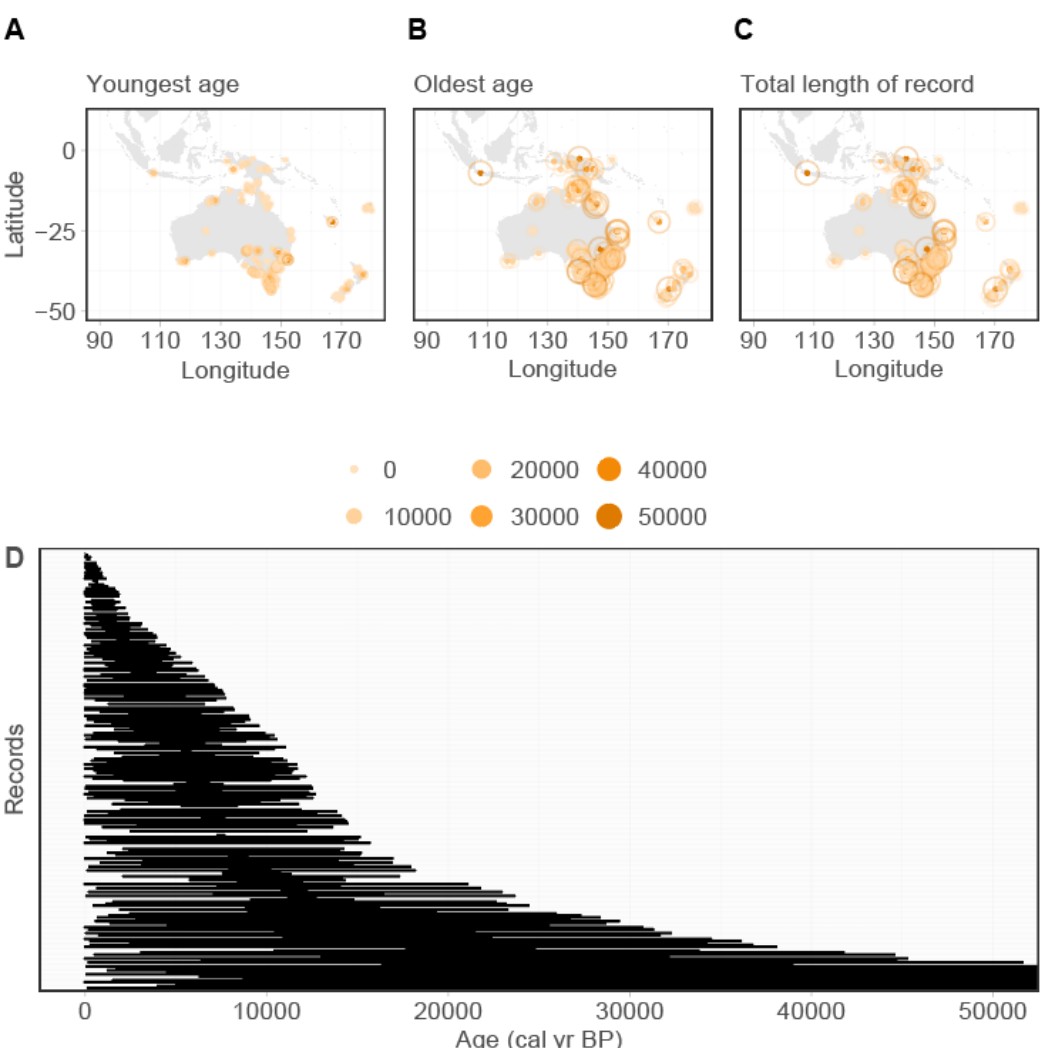

**Figure 5,** a-c) Geographical distribution of records by age, d) age distribution of the records.

### *3.4 Sedimentation rate and Deposition Time*

For 61% of the records, it takes less than 50 years to accumulate 1 cm of sediment. For 1.3% of the records, it takes 300 years or more (Fig. 6 and Appendix A, Fig. A6). The average deposition time for the IPPD is 64.8 yr/cm, but there is a very wide range, with the slowest regional sedimentation rate (largest deposition time, measured as number of years/cm), on average being from the east coast of Australia. The region with the largest range of values is arid Australia, encompassing the highest and lowest rates overall. All regions in Australia except for one (south coast) has a slower average rate than any non-Australian region, and also a much larger range (Fig. 6b). This could be due to lower average rainfall in Australia, which may inhibit biological activity in the lakes and wetlands that represent the majority of sites in the IPPD and slow down sediment accumulation. This is known to be an issue in arid and semi-arid areas (Ward and Larcombe, 2003), and with the rest of the regions represented

in the IPPD being generally high rainfall regions, this seems a likely explanation for the differences in rates. The average annual rainfall across all Australian IPPD sites is 1010 mm yr$^{-1}$, and across all non-Australian IPPD sites 2505 mm yr$^{-1}$. The reason Australian regions have larger ranges is probably due to the number of sites represented, with most sites in the database being Australian. The least represented Australian region, the southwest, also has the smallest range of any Australian region.

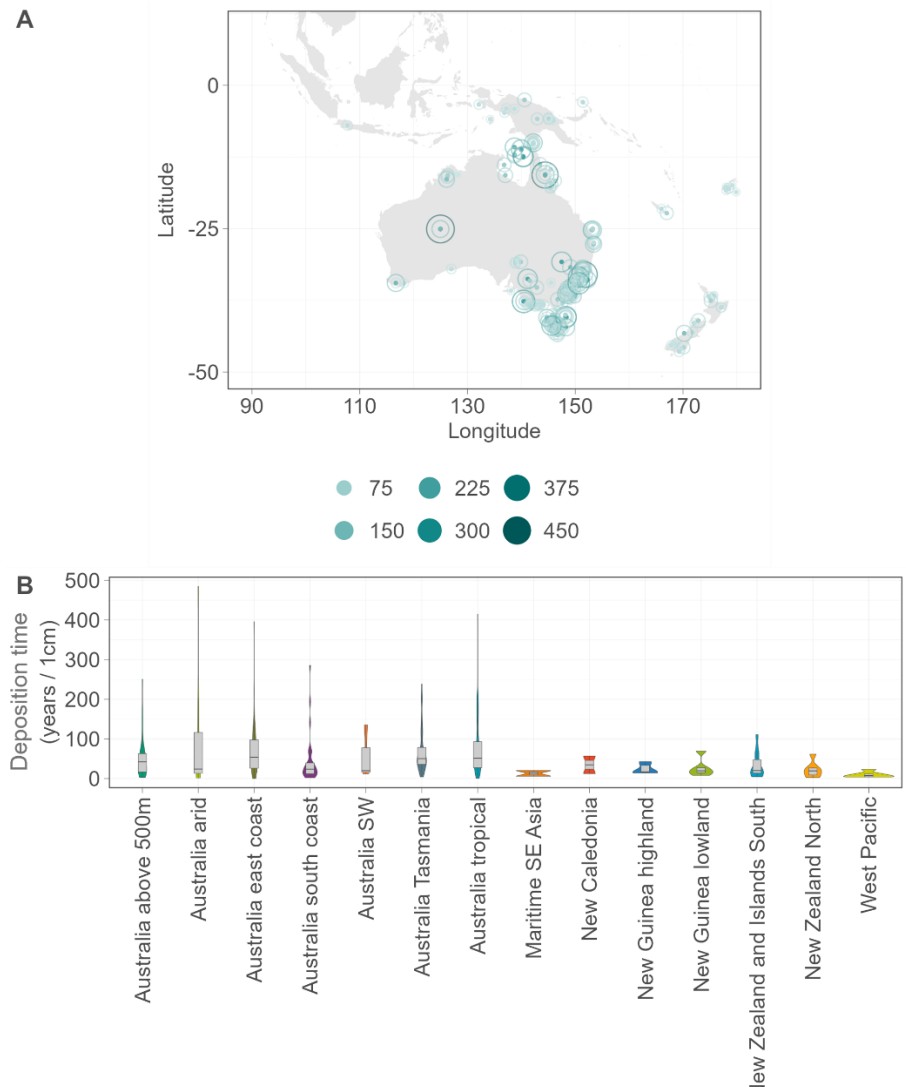

**Figure 6,** a) Geographical distribution of records by deposition time; b) deposition time by geographical region, with the mean marked by a line. For a detailed histogram of the number of records by deposition time, see Fig. A6 in the Appendix.

A large study of sedimentation rates in eastern North America found that on average mid-latitude (40-50°N) sites accumulated sediments faster (low number of years/cm) than low- or

high-latitude sites (Webb and Webb, 1988). The authors attributed the rate in low-latitude sites to the fact that most of these sites were shallow and affected by dry seasons, making them prone to periods of erosion. This could certainly also be the case for many of the Australian sites, as it is common for lakes here to fluctuate in size or to periodically dry out completely. A more recent study focusing on the northeastern United States found that deposition times are more significantly related to depositional environment, sediment age and depth than to latitude, longitude or altitude (Goring *et al.,* 2012). Here we have not examined the regional deposition times in that much detail, and it is therefore possible that the results differ between the regions due to the different depositional environments represented (Fig. 3).

## 4. Conclusions and future work

The Indo-Pacific region has been poorly represented in global repositories in the past (Gajewski, 2008). Having this resource available through Neotoma for the first time will go some way to create a truly global pollen database, and full integration of the IPPD into Neotoma complements other efforts focusing on getting more datasets into Neotoma from Africa (Lézine et al., 2021) and South America (Flantua et al., 2015). Neotoma operates with a constituent database structure (Williams et al. 2018) and sites within the IPPD can be viewed through Neotoma's informatics ecosystem (e.g., through the Explorer app (https://apps.neotomadb.org/explorer/) or using the *neotoma* R package (Dominguez Vidaña and Goring 2023)). Going forward, Annika Herbert will serve as lead steward for the IPPD within Neotoma. Possible future analyses of our database include the examination of human impact on regional vegetation, contrasting first human arrival and colonisation (e.g. López-Sáez et al., 2014; Flantua et al., 2016), examining human cultures and food production based on anthropogenic indicators (e.g. Flantua and Hooghiemstra, 2023) or the assessment of rates of vegetation change during the Holocene (e.g. Mottl et al., 2021). With such a large database now publicly accessible and open access workflows to process and standardise large compilations (Flantua et al., 2023; Vidaña and Goring, 2023), countless opportunities for further study are available, including global and hemispheric syntheses.

Considerable work has already been done using the IPPD or similar compilations. This work includes Rate of Change analysis and rainfall seasonality reconstructions on the Australian part of the IPPD going back to the last glacial period (Cadd *et al.,* 2021; Herbert and Fitchett, 2021) and Holocene plant trait analysis for the southeastern Australian part of the database

(Adeleye *et al.,* 2023). The latter is similar to a study conducted on European pollen samples by Veeken *et al.* (2022) as well as several other similar studies (e.g. van der Sande *et al.,* 2019; Lacourse and Adeleye, 2022). This highlights the importance of regional coverage, as

it can be used to perform comparative studies and examine differences or similarities with the rest of the world. Another example of this is the work by Mariani and colleagues (Mariani *et al.,* 2016, 2017, 2022), using the pollen-to-vegetation conversion model REVEALS (Sugita, 2007a, b), which has been widely used in Europe and North America for the past decade or so (Gaillard *et al.,* 2010; Sugita *et al.,* 2010). Mariani's work represents the first use of this

valuable model in Australia, and it has been instrumental in shedding light on Indigenous fire management practices and their importance (Mariani *et al.,* 2022).

Future work using the IPPD could likewise take inspiration from methodologies employed in Europe and North America to conduct similar research in the Indo-Pacific region or use it to complete a global synthesis of a commonly used technique. Examples of the latter include

large-scale quantitative climate reconstructions using various statistical techniques. Such studies are not commonly performed in the Indo-Pacific region (but see Cook and van der Kaars, 2006), but the potential has been proven previously (Herbert and Harrison, 2016). Another possibility is to perform an in-depth study of deposition times, examining the influence of factors such as latitude, altitude, depositional environment, sediment age and

depth, similar to studies in the United States (Webb and Webb, 1988; Goring *et al.,* 2012). Fully accessible global palaeoecological databases make these types of studies possible, and with more sites being added every day, the possibilities for innovative research will likewise expand.

**Acknowledgements**

The late Professors Geoff Hope and Eric Grimm were integral in the early stages of this project as well as in creating earlier versions of the database and initial upload efforts. None of the current work would be possible without their efforts. Data were obtained from the Neotoma Paleoecology Database (http://www.neotomadb.org) and the IPPD constituent

database. The work of data contributors, data stewards, and the IPPD / Neotoma community is gratefully acknowledged. The assistance of Matthew Jacques, Matthew Langer, Michael Rehani, Grace Roper, and Jocelyn Wai-Yee Lam in preparing IPPD records for upload as undergraduate students at University of Wisconsin–Madison is gratefully acknowledged.

**FUNDING**

AVH is supported by the Australian Research Council's Centre of Excellence in Biodiversity and Heritage. OM and SGAF have been supported by the European Research Council (ERC) under the European Union's Horizon 2020 research and innovation programme (grant agreement No. 741413) to the project called 'HOPE Humans On Planet Earth—Long-term Impacts on biosphere dynamics'. Additionally, SGAF acknowledges support from Trond

Mohn Research Foundation (TMF) and the University of Bergen for the startup grant 'TMS2022STG03' to the project called 'Past, Present, and Future of Alpine Biomes Worldwide - PPF-Alpine'. OM has been supported by the Czech Science Foundation PIF grant (23-063861), by the Charles University Research Centre program (UNCE/24/SCI/006), and by the Institutional Support for Science and Research of the Ministry of Education,

Youth and Sports of the Czech Republic. JLB was supported by the U.S. National Science Foundation Division of Earth Sciences (NSF EAR) 1948579.

**DATA AVAILABILITY**

The code for all figures is available at our public Github repo here; https://github.com/HOPE-
UIB-BIO/IPPD_overview. All data were processed in R using Fossilpol (Flantua et al., 2023). Over half the sites are freely available through Neotoma, the rest are in the process of being uploaded, to be completed by April 2025.

**AUTHOR CONTRIBUTIONS**

AH, SG, SF and OM conceptualised the paper; AH wrote the first draft and performed data analysis; OM made the figures; all authors except GH helped with data processing and reviewed drafts. GH compiled the first database and initialised the project.

**COMPETING INTERESTS**

The authors declare that they have no conflict of interest.

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

**Appendix A**

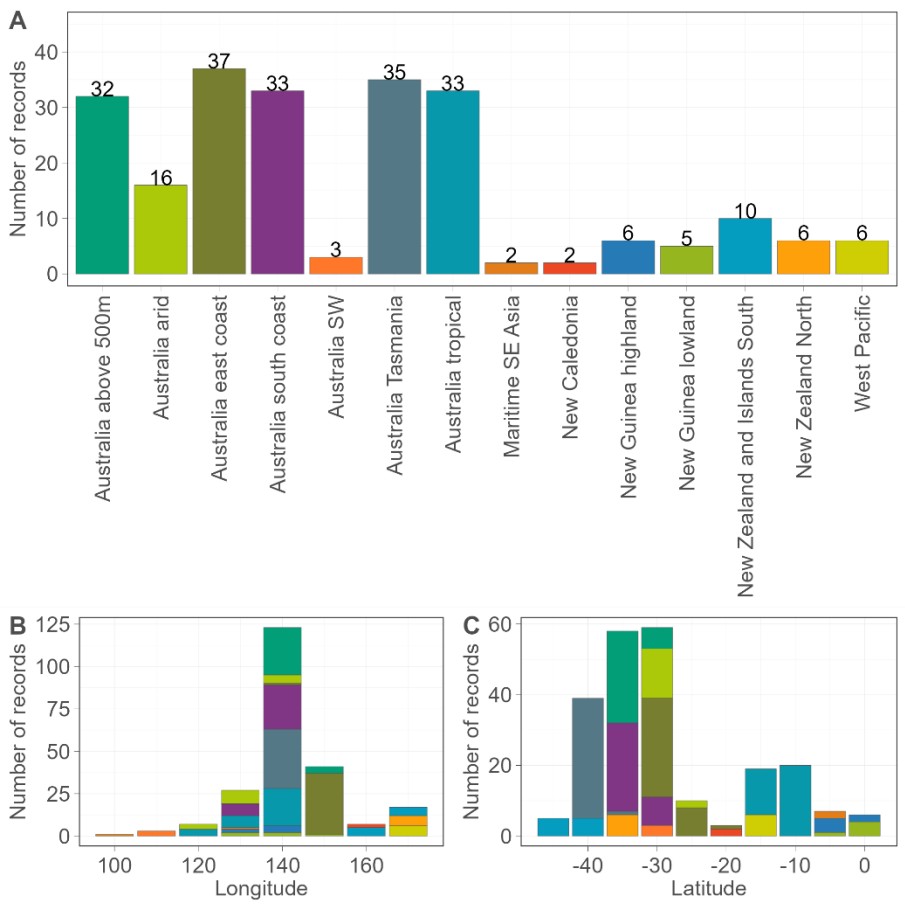

**Figure A1,** a) Number of records by geographical region; b) Number of records by latitude, colours representing the regions from a; c) Number of records by longitude, colours representing the regions from a.

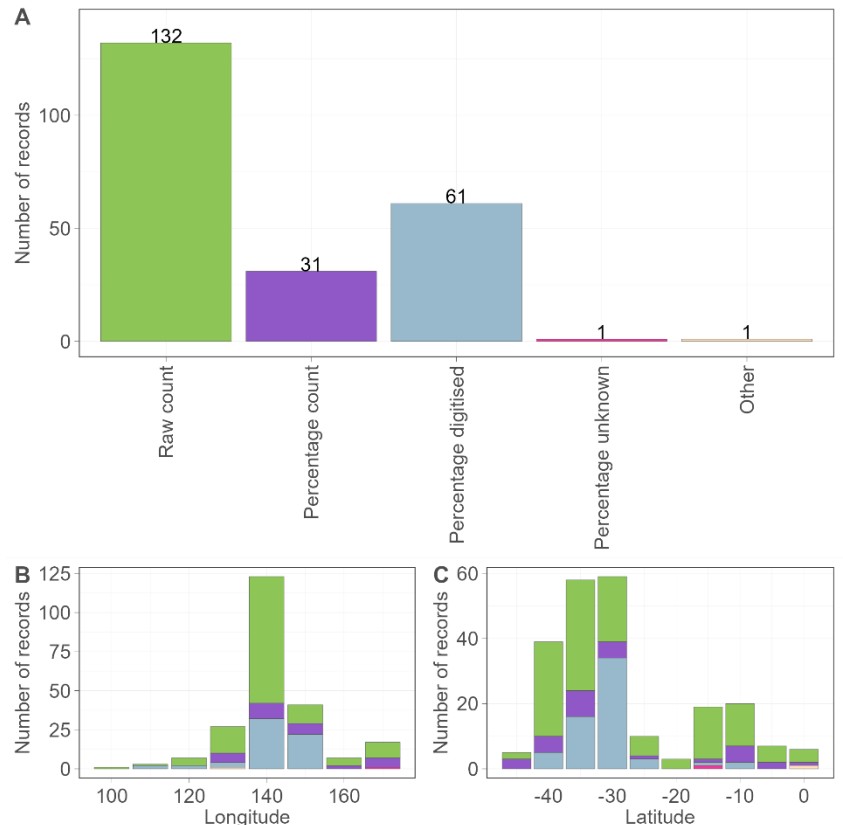

**Figure A2,** a) Number of records by count type; b) Number of records by latitude, colours representing count type from a; c) Number of records by longitude, colours representing count type from a.

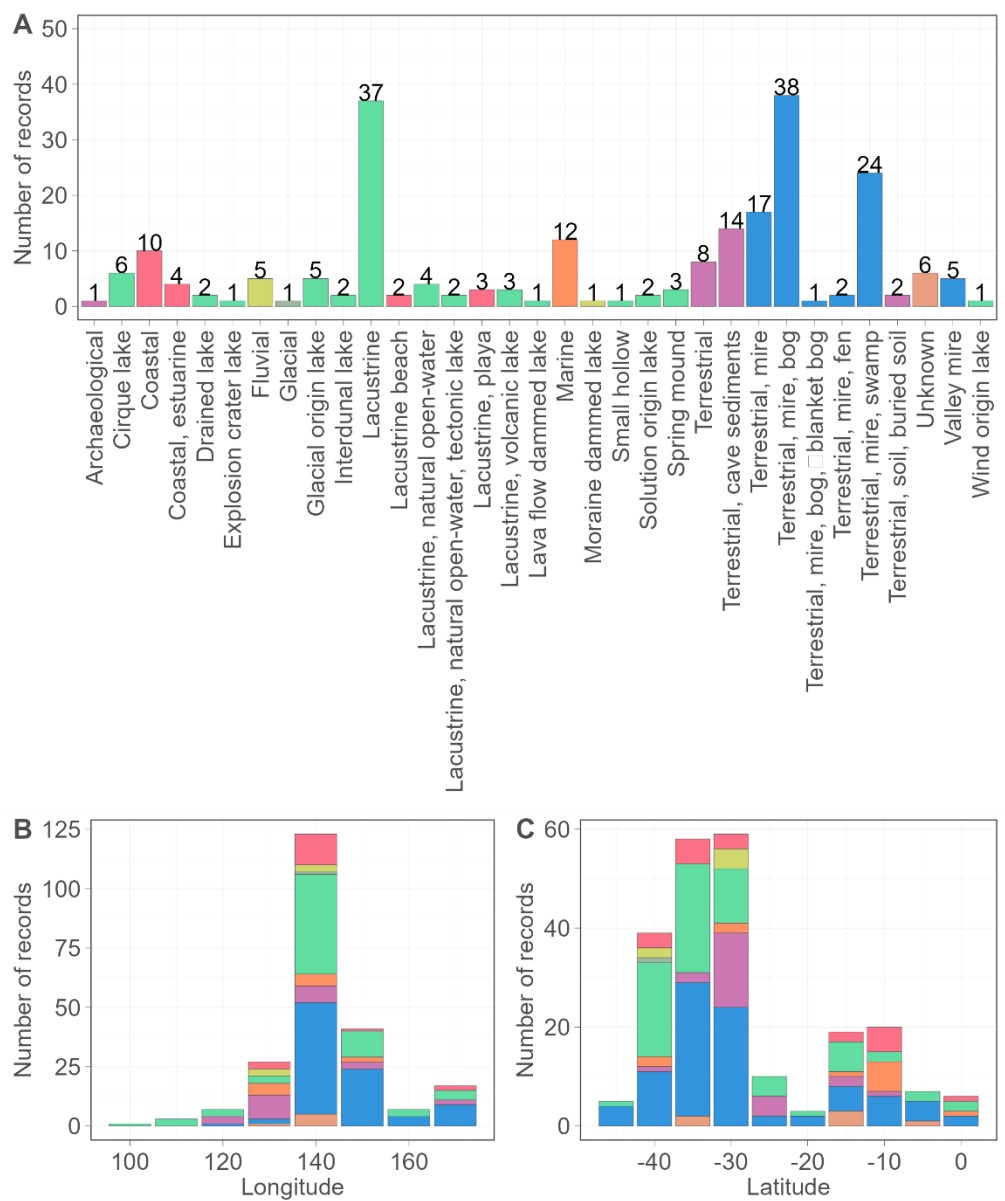

**Figure A3,** a) Number of records by sedimentary environment; b) Number of records by latitude, colours representing sedimentary environment from a; c) Number of records by longitude, colours representing sedimentary environment from a.

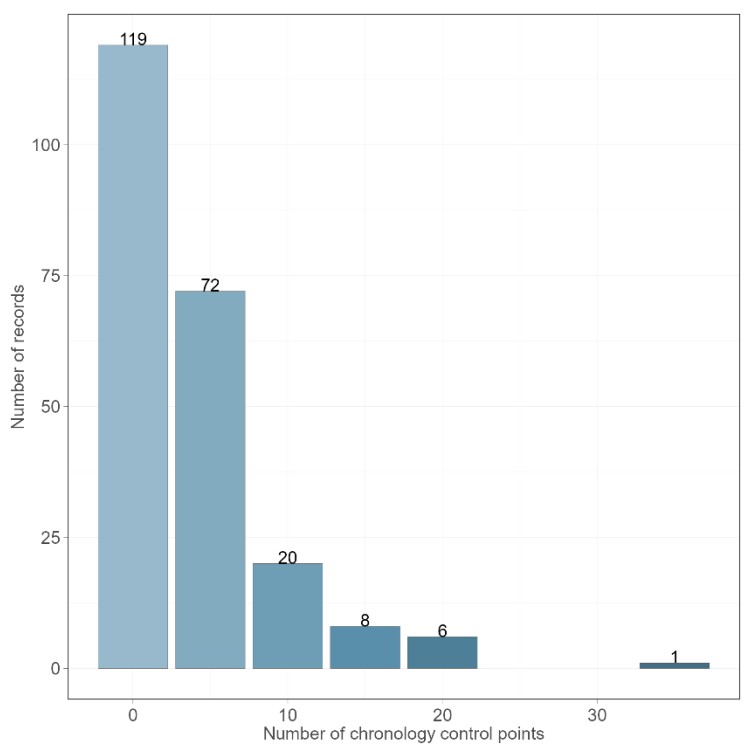

**Figure A4,** Number of records by number of chronological control points.

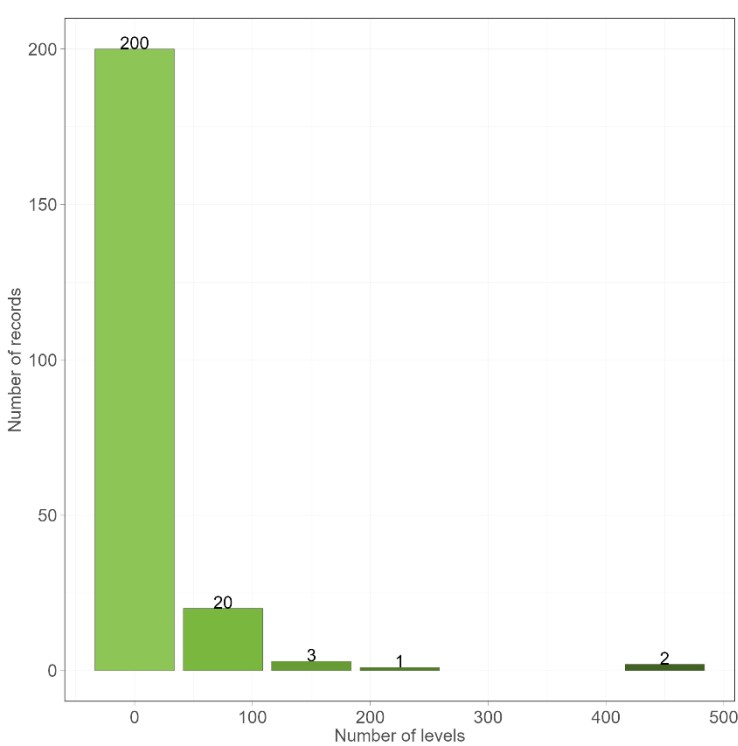

**Figure A5,** Number of records by number of levels.

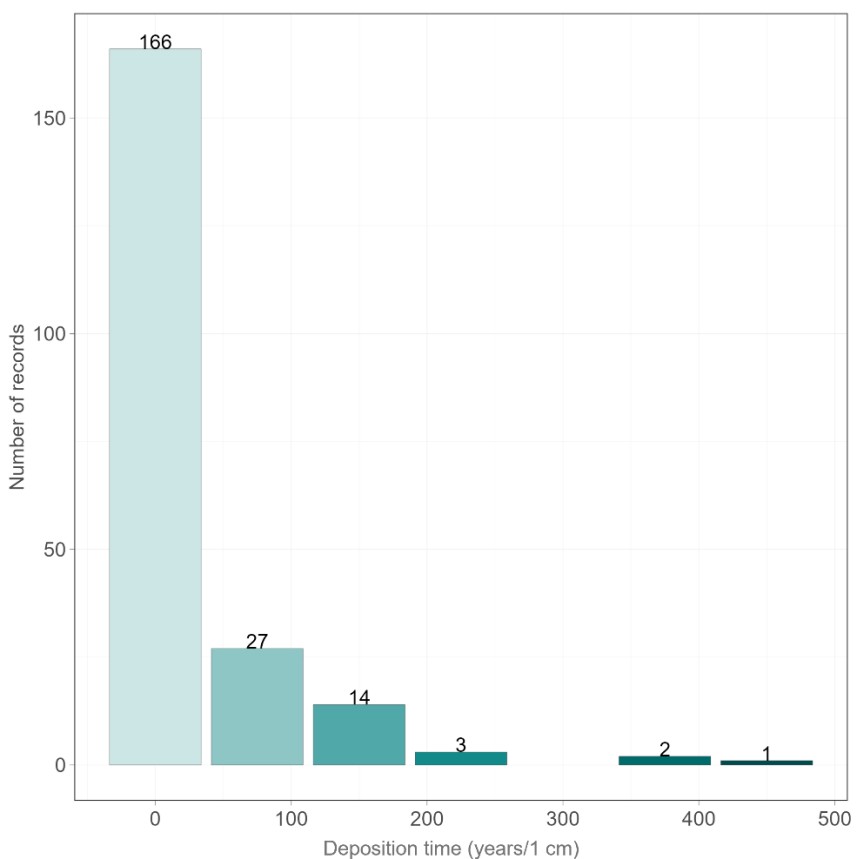

**Figure A6,** Number of records by deposition time.