# Peer review of "The Indo-Pacific Pollen Database - a Neotoma constituent database"

_Climate of the Past, 2024_

## Author Response (AR1)

We would like to thank the reviewers for their comments and attention to detail in their reviews.

*Reviewer 1:* The authors' study on the Neotoma's Indo-Pacific Pollen Database (IPPD) is an important work that would serve as a link between data users/researchers and the Neotoma Paleoecology Database, meaning that it will make IPPD easier to find through search engines. While the work is well structured and presented, the aspect of the application of the database can be better developed to capture the range of studies/investigations the database can be used for. There is currently an emphasis on rates of vegetation change (RoC), which is only one feature of vegetation, there are others that have/can be investigated. Also, there is an emphasis on Northern Hemisphere work with no acknowledgement of existing studies in Southern Hemisphere area of study (Indo-Pacific region).

**Ln 34-37:** Consider rewording to something like, 'The IPPD offers many exciting research opportunities to investigate Holocene regional vegetation changes and associated drivers, including contrasting the impact of first human arrival and European colonization on vegetation'. Alternatively, you can include other examples of research themes to the RoC, such as floristic diversity (alpha diversity), land cover reconstruction, functional/trait diversity etc. For more clarity, it will be helpful to also let the readers know somewhere in the introduction that rate of vegetation change is also referred to as temporal compositional turnover or beta diversity.

**Response:** This has been rephrased to: "The IPPD offers many exciting research opportunities to investigate past regional vegetation changes and associated drivers, including contrasting the impact of first human arrival and European colonisation on vegetation. Examining spatio-temporal patterns of diversity and compositional turnover/rate of change, land cover reconstructions, plant functional or trait diversity are other avenues of potential research, amongst many others."

**Ln 51-53**: Consider including 1-2 example study references where database-based study/meta-analysis of ecological histories have been used to predict future changes.

**Response:** A reference has been added.

**Ln 69-73:** The reference provided here gives the impression that no IPPD-based study focused on the Indo-Pacific area has ever been done, which is not the case, there is a number of IPPD-based study in the region. An example is Mariani et al. 2020 https://doi.org/10.1002/fee.2395 and a number of others. Peter Kershaw possibly also used the earlier version of the database to conduct this study https://doi.org/10.1016/0034-6667(94)90021-3 in the early 90s. Strandberg et al. 2024 did some work in the Pacific Islands https://www.nature.com/articles/s41559-023-02306-3. Nogue et al 2021 global meta-analysis work captures the Pacific Islands as well https://doi.org/10.1126/science.abd6706 . Capturing all this range of studies is important to convey what can be done using the database, and this is not captured with the two studies currently cited. Furthermore, the Southern Hemisphere is generally underrepresented in this space and the current Indo-Pacific focused paper should be an opportunity to showcase studies from the Southern Hemisphere region/regions.

**Response:** This paragraph was intended as background to Neotoma only but has been expanded to include a section on regional studies, those using the IPPD, as well as others. We

have also included a statement on the general representation of Southern Hemisphere data in Neotoma.

**Figure 2 caption:** Consider spelling out MAT and MAP for clarity.

*Response:* All climate variable abbreviations have now been spelled out in the caption.

**Conclusion:** Again, the proposed future work implies no study on rates of change or human impact using the IPPD and focused on the Indo-Pacific has been conducted, which is not the case. Consider revising here (and relevant areas above) by acknowledging existing work in the study region and then suggest areas that can be further developed. Also Veekeen et al 2022 (https://doi.org/10.1111/ele.14063) is an another example of what the database can generally be used for (function trait analysis) which should also be cited somewhere, and I believe a similar work has also been conducted in Australia.

**Response:** A paragraph on regional studies and how they relate to, and can be used to complement, global studies, has been added to the conclusion section.

*Reviewer 2:* The accessible databases are key to understanding ecological histories, climate changes and its potential forcing mechanisms, thus, helpful for predicting future environmental changes in global warming. In this study, Herbert et al. comprises 226 fossil pollen records since the last glacial period in the Indo-Pacific Pollen Database (IPPD), and by integrating the IPPD into the online Neotoma Palaeoecology Database. It fulfills the gap for the global pollen syntheses, and is useful for better understanding ecological changes during the Quaternary period.

In current version, I would suggest a minor revision before accepting it for publication. Here are a few basic comments that could guide the authors to submit a more detailed manuscript.

1. In the methods section, what is the criteria for the digitized pollen data from publications or theses?

**Response:** An explanation of the digitisation quality control procedure has now been added to the method section.

2. Lines 112-115: the percent of raw counts in the IPPD is 58.4%, and 27% for digitized data, why the total value is not reach 100%? And some other data in the IPPD?

**Response:** An explanation has been added.

3. In Fig.1, what is the percentage unknown and other?

**Response:** An explanation of these categories has been added to the caption.

4. Line 133: a total of 33 different depositional environments represented in the database, but the number of the depositional environments is not 33 types in the Fig A3, please check it.

**Response:** This was due to a database error which has been rectified, we would like to thank the reviewer for spotting this unfortunate oversight. The figure has been corrected.